# Electrically stimulated eccentric contraction during non-weight bearing knee bending exercise in the supine position increases oxygen uptake: A randomized, controlled, exploratory crossover trial

**Hiroshi Tajima, Hiroo Matsuse**◯*, **Ryuki Hashida, Takeshi Nago, Masafumi Bekki, Sohei Iwanaga, Eriko Higashi, Naoto Shiba**

Rehabilitation Center, Kurume University, Kurume, Fukuoka, Japan

* matsuse_hiroh@kurume-u.ac.jp

## Abstract

It is well known that prolonged bed rest induces muscle weakness, muscle atrophy, cardiovascular deconditioning, bone loss, a loss of functional capacity, and the development of insulin resistance. Neuromuscular electrical stimulation is anticipated to be an interventional strategy for disuse due to bed rest. A hybrid training system (HTS), synchronized neuromuscular electrical stimulation for voluntary exercise using an articular motion sensor, may increase the exercise load though bed rest. We assessed oxygen uptake or heart rate during knee bending exercise in the supine position on a bed both simultaneously combined with HTS and without HTS to evaluate exercise intensity on different days in ten healthy subjects (8 men and 2 women) by a randomized controlled crossover trial. The values of relative oxygen uptake during knee bending exercise with HTS were significantly greater than those during knee bending exercise without HTS (7.29 ± 0.91 ml/kg/min vs. 8.29 ± 1.06 ml/kg/min; p = 0.0115). That increment with HTS was a mean of 14.42 ± 13.99%. Metabolic equivalents during knee bending exercise with HTS and without HTS were 2.08 ± 0.26 and 2.39 ± 0.30, respectively. The values of heart rate during knee bending exercise with HTS were significantly greater than those during knee bending exercise without HTS (80.82 ± 9.19 bpm vs. 86.36 ± 5.50 bpm; p = 0.0153). HTS could increase exercise load during knee bending exercise which is easy to implement on a bed. HTS might be a useful technique as a countermeasure against the disuse due to bed rest, for example during acute care or the quarantine for infection prophylaxis.

## Introduction

It is well known that muscle atrophy, cardiovascular deconditioning, bone loss, a loss of functional capacity, and the development of insulin resistance due to disuse occurs in astronauts [1, 2]. Bed rest causes similar disuse [3] although bed rest has been prescribed in the past for several other clinical conditions such as the rest for the intensive care or the quarantine for

**Data Availability Statement:** All relevant data are within the paper and its Supporting Information files.

**Funding:** The authors received no specific funding for this work.

**Competing interests:** The authors have declared that no competing interests exist.

**Abbreviations:** ES, electrical stimulation; HTS, hybrid training system; HR, heart rate; NMES, neuromuscular electrical stimulation; METs, Metabolic equivalents; RER, respiratory exchange ratio; VC, volitional contraction; VCO2, carbon dioxide output; VO2, oxygen uptake.

prevention of infection. Strength training or aerobic exercise programs are expected to prevent disuse atrophy and deconditioning [4]. However, the evidence regarding interventions involving either strength training alone or aerobic exercise alone remains uncertain [4]. Moreover, some patients in acute care hospitals have difficulty even standing. Some patients aren't able to do exercise at intensity 3 Metabolic equivalents (METs) a such as walking. We need relatively efficient and simple exercise that can be performed on a bed to prevent disuse during bed rest.

Neuromuscular electrical stimulation (NMES) is widely used to lessen the complications of disuse during spaceflight or bed rest by increasing muscle strength and mass [5–7]. Moreover, it is well known that NMES upregulates glucose metabolism and peak oxygen uptake [8, 9]. In addition, the combined application of electrical stimulation (ES) and volitional contractions (VC) (performing ES during VC either non-simultaneously or simultaneously) is said to be more effective than ES or VC alone [10]. The hybrid training system (HTS) is one of the techniques that simultaneously combines VC and ES, and is able to be used during exercises such as ergometer walking. HTS utilizes electrically stimulated eccentric contractions. Eccentric contractions are important for muscle strengthening [11]. Furthermore, electrically stimulated eccentric contractions are recognized as a muscle strengthening technique, because we can obtain bigger muscle torque (exercise load) than with concentric contractions alone [12, 13]. It has been reported that HTS can increase both muscle strength and mass even when exercise intensity is relatively low [14–17]. Moreover, we have reported that ergometer exercise using HTS leads to a greater increase in metabolic cost [18, 19]. Knee bending exercise, one of the common early mobilizations on a bed, is recommended for critically ill patients [20]. Therefore, we hypothesize that by combining HTS with knee bending exercise may provide exercise intensity that can easily be performed on a bed. It may be a useful exercise technique for the prevention of disuse due to bed rest.

The purpose of the present study is to compare the metabolic cost between knee bending exercise on a bed with and without HTS by analyzing expired gas.

## Methods

### Subjects

This study protocol was approved by the Ethics Committee of Kurume University (No.19209) and registered at UMIN Clinical Trials Registry (UMIN000039336). Following approval, written informed consent was obtained from 10 healthy (8 men and 2 women, age 28.9 years (SD = 8.5) who had reviewed the goals of the study and agreed to participate. The subjects of this study were healthy participants as preliminary research before intending for the clinical patients. When participants were enrolled, the staff who was not involved in intervention and evaluation randomized the intervention sequence using a random number generator and sealed opaque envelopes with a block size of 5 to allocate to group A (the knee bending exercise test with HTS was performed firstly) or group B (the knee bending exercise test without HTS was performed firstly) (Fig 1). Inclusion criteria were applied: age between 20–50 years, non-smokers, body mass index less than 30, doing normal physical activity, and normal physical function. A specialist in expert orthopedics and rehabilitation judged the normal physical function from the physical findings such as muscle strength, sensation, and range of motion according to the criteria of the Japanese Orthopedic Association. Exclusion criteria were that the subjects had: a musculoskeletal problem, adverse medical history affecting cardiorespiratory function, or some kind of medical treatment. We set these criteria in order to minimize possible confounding effects of subject characteristics over the exercise tests in this study. They were also not to participate in any regular sports activities during the study period.

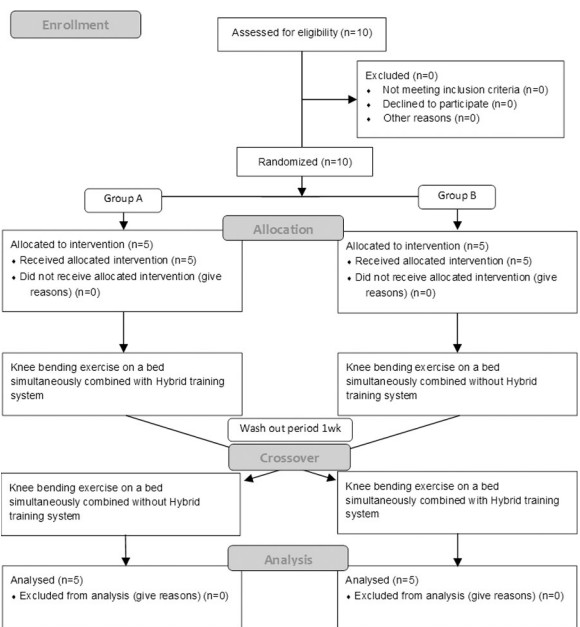

**Fig 1. Flow diagram.** Subjects were randomly allocated to group A (the knee bending exercise test with Hybrid training system was performed firstly) or group B (the knee bending exercise test without Hybrid training system was performed firstly) and then transitioned to the alternative exercise test on different days separated by an interval of one week as a wash out period.

## Design

We performed two cardiopulmonary exercise tests in the clinical laboratory of the rehabilitation center by a randomized, controlled, crossover trial (Fig 1). It is very difficult for both subject and researcher to completely blind presence or absence of electrical stimulation. Moreover, we need to minimize the influence of the individual difference. Therefore, we chose a crossover trial in this study. Environmental conditions were similar for all exercise tests (21 to 24 degrees centigrade, 45 to 55% relative humidity). We measured gas exchange while subjects did knee bending exercises with a RISTA board (NIPPON SIGMAX Co, Tokyo, Japan) at the lowest exercise resistance load using the following protocol after a rest for 30 minutes. The RISTA board is a Japanese medical device to facilitate knee bending exercises in the supine position on a bed. We used the device to achieve a stable knee bending exercise on a flat surface. During the tests, gas exchange data was collected continuously with an automated breath by breath system (AE-100i, Minato Medical Science Co. Ltd., Osaka, Japan) using the standard technique. The AE-100i consists of a microcomputer, a hot wire flowmeter, and a gas analyzer, which contains a sampling tube, filter, suction pump, oxygen ($O_2$) analyzer made by a paramagnetic $O_2$ transducer, and an infrared carbon dioxide ($CO_2$) analyzer. Ventilatory parameters were measured using a hot-wire flow meter, and the flow meter was calibrated with a syringe of known volume (2.0l). A zirconium sensor and an infrared absorption analyzer, respectively, measured O2 and CO2 concentrations. The gas analyzer was calibrated to known standard gas levels (O2 15.16%, CO2 5.023%) before each test. The intraclass correlation coefficients using Case 2 (2,1) for VO2 of the primary outcome was 0.876 in our clinical laboratory. Also, heart rates (HR, beats/min) were continuously monitored by electrocardiogram during the tests. O2 and CO2 were calculated and recorded during the following exercise tests.

## Intervention

The knee bending exercise test was performed as two tests (with HTS or without HTS) for analysis of expired gas. Participants rested for 30 min in the supine position to minimize the influence of physical activity before the exercise test. At first subjects rested for five minutes in a sitting position to evaluate of the basic inactivity quiet metabolic cost while sitting quietly. Then, participants performed the knee bending exercise tests with or without HTS according to a sequence assigned before the exercise test. Next, participants transitioned to the alternative exercise test on different days separated by an interval of one week as a wash out period. Therefore, the participant who first performed the exercise test with HTS performed the exercise test without HTS next. In the interventional exercise, subjects performed the knee bending exercise with reciprocal 2-second (45degree/sec) knee flexion and extension contractions at the speed of 1Hz using a metronome for 5 minutes with HTS or without HTS in the supine position on a flat surface after warming up at a slow speed for 3 minutes (Fig 2). Successively, subjects performed 2 minutes cool down. Therefore, they had the total exercise time of 10 minutes in each of two tests in consideration of muscle fatigue. The knee joint range of motion was set at a nearly 90˚ arc that extended from 20˚ to 110˚ (0˚ indicating full knee extension). At the onset of constant-load exercise, O2 increases in healthy individual's mono-exponentially with a time constant to achieve a steady state below the lactate threshold within about 3 minutes [21]. Therefore, we averaged the data of HR and expired gas during the last 2 minutes of the interventional exercise time of 5 minutes of each test and used them for data analysis.

**HTS protocol.** HTS was combined with knee bending exercise in the supine position on a flat surface with an electrical stimulator (HIZA TRAINER, EU-JLM50S, Panasonic Corporation, 1006 Ohaza-Kadoma, Kadoma City, Osaka, Japan). This provided a constant voltage stimulus to the skin electrodes (regulated voltage)". This stimulator is household medical equipment including a wrapping band to fix the device and acceleration sensors or electrodes during exercise. The acceleration sensors act as joint motion sensors (EWTS9PD, Home Appliances Development Center Corporate Engineering Division, Appliances Company Panasonic Corporation 2-3-1-2 Noji-higashi, Kusatsu City, Shiga, Japan) and are placed on the front of each leg 88 mm above the patellar edge. The sensors analyze the algorithm of each gait pattern, and stimulate the flexor or extensor in accordance with the motion of each bilateral

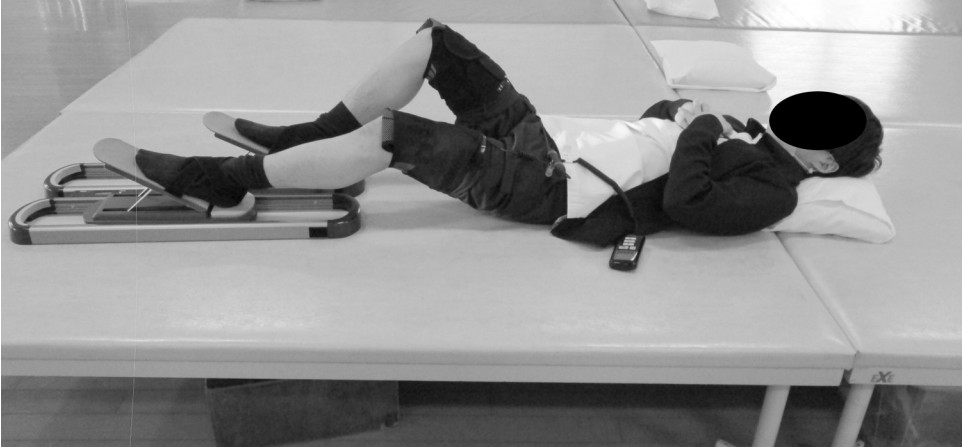

**Fig 2. Exercise scene.** Subjects lay in a supine position on a flat surface with their quadriceps electrically stimulated as they attempted to bend their knee and their hamstrings electrically stimulated as they attempted to extend their knee. The timing of the electrical stimulation was controlled by a joint motion sensor attached to the knee. They bent and stretched their left and right lower limbs alternately.

knee joint during knee bending exercise (Fig 2). Electrodes (Sekisui Plastics Co., Tokyo, Japan) coated with an oxidation-resistant silver-carbon compound and were placed over the quadriceps on the anterior thigh, and over the hamstrings on the posterior thigh with low impedance gel. The size of the electrodes was 15cm × 5cm for the quadriceps and 10cm × 6cm for the hamstrings. ES stimulated the quadriceps during knee flexion. Conversely, ES stimulated the hamstrings during knee extension. Therefore, the electrically stimulated muscles were eccentric contractions. ES parameters were based on a standard Russian waveform [22] in which a 5,000 Hz carrier frequency is modulated at 40 Hz (2.4 ms on, 22.6 ms off) to deliver a rectangular voltage biphasic pulse [16–19, 23, 24]. ES intensity was set to approximately 80% of the subject's maximum tolerance. This intensity has been reported to successfully improve muscle strength and mass without causing pain or numbness [16, 17, 23].

## Statistical analysis

All variables are presented as means ± SD. This study was not set to the sample size for an exploratory study. The original study plan recruited 20 participants, but we analyzed this study because we were able to achieve this study purpose from ten participants. We compared the participants' characteristics and rest values (relative VO2, VCO2, and HR) between groups (two sequences) by Student's t-test, or Fisher's exact test. Values for relative VO2, VCO2, HR, and respiratory exchange ratio (RER) during exercise test were analyzed with linear mixed model that included fixed effects of treatment (with HTS or without HTS), period, and sequence. Then, we set subjects(sequence) as the covariance parameter for the random effect. Also, we tried the linear mixed model in 3 covariance structures (compound symmetry, unstructured, and autoregressive). The compound symmetry was the best covariance structure as a result that it was judged in BIC. Therefore, we selected the compound symmetry as the underlying covariance structure used in the linear mixed model. Next, values for relative VO2, VCO2, and HR were assessed using a paired Student's t-test as a post hoc test in order to compare the differences between the knee bending exercise tests with and without HTS for the main purpose of this study. Moreover, we calculated the effect size (r) to know the strength of association between intervention and change on VO2 of the primary outcome. All the statistical analyses were performed using JMP Version 14.0 statistical software (SAS Institute Inc., Cary, NC, USA) and p values < 0.05 were considered to be statistically significant.

## Results

There were not any problems and adverse events in the exercise tests for subjects such as not being able to continue from pain or fatigue. Each five subjects participated in two sequences to be shown in the flow diagram (Fig 1), and all ten subjects finished all tests and were analyzed. The primary data for this study is available as S1 Table.

There were no significant differences at participants' characteristics and rest values (relative VO2, VCO2, and HR) between groups (Table 1). There were no significant differences at period and sequence in values for relative VO2, VCO2, and HR during exercise test though there was significant difference at treatment (with HTS or without HTS). As a primary outcome, the values of relative VO2 during knee bending exercise with HTS were significantly greater than those during knee bending exercise without HTS (7.29 ± 0.91 ml/kg/min vs 8.29 ± 1.06 ml/kg/min; p = 0.0115) (Fig 3). Mean within subject difference (95% confidence interval) in the values of relative VO2 between treatments (with HTS and without HTS) was 0.998 (0.3170–1.6785). The effect size was0.76. The increment using HTS was a mean of 14.42 ± 13.99%. Metabolic equivalents during knee bending exercise with HTS and without HTS were 2.08 ± 0.26 and 2.39 ± 0.30, respectively. The values of relative VCO2 during knee

**Table 1. Participants' characteristics.**

| Characteristics | Group A | Group B | p-value |
|---|---|---|---|
| Age (years) | 27.6 ± 7.3 | 30.2 ± 10.3 | 0.6580 |
| Male sex, n (%) | 4 (80) | 4 (80) | 1.00 |
| Body mass (kg) | 66.1 ± 8.6 | 59.8 ± 10.2 | 0.3244 |
| Height (cm) | 167.8 ± 5.9 | 168.4 ± 13.3 | 0.9335 |
| BMI (kg/m$^2$) | 23.4 ± 1.6 | 21.1 ± 3.1 | 0.1861 |
| Rest VO$_2$ (kg/cm$^2$) | 3.85 ± 0.24 | 3.91 ± 0.66 | 0.8606 |
| Rest VCO$_2$ (kg/cm$^2$) | 3.52 ± 0.70 | 3.33 ± 0.80 | 0.6947 |
| Rest HR (bpm) | 68.6 ± 4.7 | 68.5 ± 9.4 | 0.9906 |

Mean ± SD. P-values were for comparing between groups by Student's t-test, or Fisher's exact test.

Abbreviations: BMI, body mass index; Rest HR, heart rate in a sitting position; Rest VCO2, carbon dioxide output in a sitting position; Rest VO2, oxygen uptake in a sitting position.

bending exercise with HTS were significantly greater than those during knee bending exercise without HTS (7.29 ± 0.91 ml/kg/min vs 8.29 ± 1.06 ml/kg/min; p = 0.0423) (Fig 4). There was not a significant difference between the values of RER during knee bending exercise with HTS and without HTS (0.94 ± 0.08 vs 0.94 ± 0.10; p = 0.86). The values of HR during knee bending exercise with HTS were significantly greater than without HTS (80.82 ± 9.19 bpm vs 86.36 ± 5.50 bpm; p = 0.0153) (Fig 5).

## Discussion

This is the first study of the metabolic cost of knee bending exercise combined with NMES that is easily done in the supine position. This study showed that HTS, utilizing electrically stimulated eccentric muscle contractions as a resistance to voluntary exercise, increased exercise intensity during knee bending exercise on a bed. HTS may be a useful exercise technique to prevent disuse during bed rest by providing augmentation of local mechanical stress and systemic metabolic stress.

The decrement in mechanical stress caused by bed rest or spaceflight causes muscle weakness, muscular atrophy, bone atrophy, and the loss of cardiovascular capacity. Resistance exercise is an effective countermeasure for the prevention of muscle atrophy [2] even if it is relatively low volume of resistance exercise [25, 26]. However, Bryan. R. Oates et al. [25] used an exercise intensity of 80% of maximal with resistance exercise. It is difficult for clinical patients to achieve adequate exercise intensity on a bed. The flywheel ergometer has been used

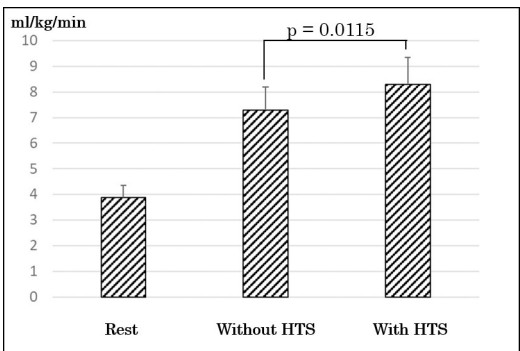

**Fig 3. Comparing of changes of oxygen uptake.** HTS, hybrid training system.

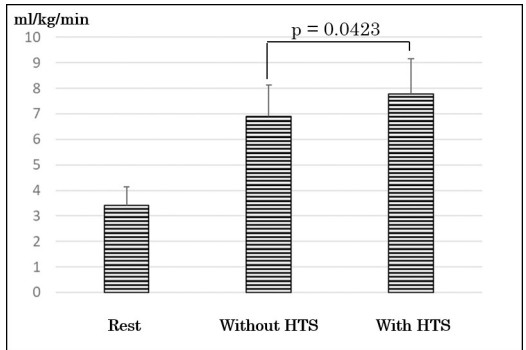

**Fig 4. Comparing of changes of carbon dioxide production.** HTS, hybrid training system.

for resistance training in space flight. The resistance was provided independent of gravity by using the inertial focus of a flywheel [27]. B. A. Alkner and P. A. Tesch [26] reported the utility of the flywheel ergometer on the bed. However, it is difficult to use a flywheel ergometer for clinical applications because it is a special device. NMES is widely used in clinic settings and can easily generate sufficient muscle contractions without relying on gravity. Therefore, NMES could be useful as a countermeasure against disuse due to bed rest. It could be especially useful when effective voluntary exercise is not possible due to disturbance of consciousness or strict medical management [28]. However, VC is undoubtedly the most basic effective countermeasure for disuse. In this study, the knee bending exercise which was easily performed in the supine position on a flat surface was approximately 2 METs in exercise intensity. Furthermore, we can make up for the detriments and use them to advantage by combining VC with ES [18, 19, 24, 29]. Moreover, ES should ideally be applied synchronously with voluntary exercise to obtain synergistic benefits [2]. In this study, we could increase exercise intensity by approximately 14.4% of the by combining ES with the knee bending exercise in the supine position on a flat surface. Therefore, we could strengthen the exercise intensity even though in the supine position on a flat surface by using HTS more effectively.

In addition, aerobic exercise is an effective countermeasure against the loss of cardiovascular capacity associated with disuse [2]. Generally, the metabolic rate during complete bed rest is less than 1 MET. Physical activity is undoubtedly the most effective countermeasure against disuse. The knee bending exercise in this study resulted in approximately 2 METs of metabolic rate. NMES stimulates metabolic response [30] and energy consumption, carbohydrate oxidation, and whole-body glucose uptake [9]. Therefore, we may use NMES as aerobic exercise to

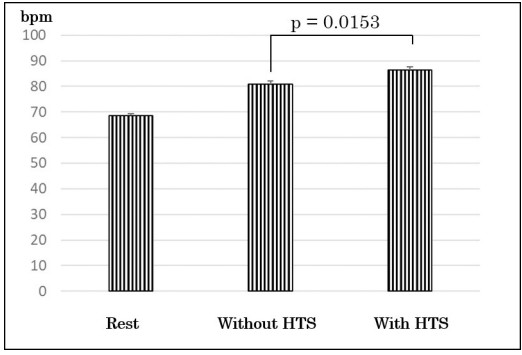

**Fig 5. Comparing of changes of heart rate.** HTS, hybrid training system.

improve physical capacity [31]. However, NMES did not change maximum capacity although aerobic exercise did in rats [32]. Additionally, it has been shown that the use of NMES is safe in patients after cardiovascular surgery [33], but NMES did not improve functional capacity in cardiac valve surgery patients in the immediate postoperative period [34]. Therefore, the aerobic exercise effect of NMES as a countermeasure against disuse due to bed rest may be limited. ES activation is considered nonselective with regard to the type of motor unit and synchrony, and it preferentially activates Type II fibers as compared with VC [35]. Jean Theurel et al. showed that the O2 and RER were significantly higher during ES compared with VC to compare respiratory gas exchange variables between equal-intensity (i.e., same force output) ES and VC of the quadriceps [36]. They considered that these findings probably reflect differences in the patterns of motor unit recruitment between ES and VC. On the other hand, in this study, there was not a significant difference between the values of RER between HTS and VC although O2 during HTS was significantly greater than for VC (without HTS). Combining VC with ES simultaneously seems to have been responsible for those results. Similarly, in a past study, we showed that HTS causes an increase of approximately 4.4% of O2 during unloaded cycling without a significant difference in RER [18]. Moreover, HTS caused an increase of approximately 21.1% of O2 during aerobic cycle ergometer exercise at moderate-intensity [19]. In this study, HTS caused an increase of approximately 14.4% of O2 during knee bending exercise in the supine position on a flat surface. In addition, the METs for these studies were approximately 2.39, which is equivalent to light effort such as housework or light exercise. Moreover, a utility of NMES was shown in disuse prevention or the complications prevention in the acute care [37]. Furthermore, NMES is expected as one of the effective exercise therapies during the quarantine for COVID-19 patients [38]. These findings showed the possibility that we might strengthen the metabolic stress from increased physical activity by combining HTS with a knee bending exercise that is easily performed by bed ridden patients (e.g., rest during the acute care or the quarantine for prevention of infection spread).

There are a few potential limitations of this study. Because this study is an exploratory study, the sample size was not set. At a significance level of 0.05, an effect size in the VO2 variable of 0.76, and a power of 0.80, to detect an inter-group difference in means of 0.998. Nm would require a minimum of 58 subjects for a two-treatment parallel-design study. The purpose of this study was to evaluate the basic gas exchange response to HTS combined with knee bending exercise as a technique to provide physical activity as a countermeasure against disuse due to bed rest. For estimating exercise intensity, oxygen uptake reserve or heart rate reserve is preferable [39]. However, we did not evaluate peak oxygen consumption or peak heart rate in each subject. In addition, we did not evaluate lactic acid or glucose to evaluate the metabolic response after having done strict nutritional management. For the investigation of clinical response, it is necessary to include clinical patients. Furthermore, the group effect (sequence) was absent, but it is unclear whether we could wash out the influence of HTS in one week. Therefore, a study not implementing a crossover trial may be necessary. Additionally, muscle contraction quantity and time influence kinetic oxygen uptake. Therefore, various protocols of electrical stimulation (such as modification of frequency, pulse width, or duty cycle) will be necessary to evaluate the effect of HTS in a future study. Of course, we should perform a randomized, controlled trial design, long-term investigation study to investigate the efficacy of strength and cardiovascular capacity of HTS for patients with disuse.

## Conclusion

These findings indicate that the combined application of VC and EC could lead to a greater increase in metabolic cost during knee bending exercise in the supine position on a flat surface.

Knee bending exercise simultaneously combined with HTS might be useful as a countermeasure against disuse due to bed rest by enforcing metabolic stress as well as mechanical stress.

## Supporting information

**S1 Table. Primary data of the study.**
(XLSX)

**S1 Checklist. CONSORT 2010 checklist of information to include when reporting a randomised trial***.
(DOC)

**S1 File.**
(DOCX)

**S2 File.**
(DOCX)

**S1 Data.**
(PDF)

## Acknowledgments

We would like to thank Kurume rehabilitation center for supporting the expired gas measurements.

## Author Contributions

**Conceptualization:** Hiroo Matsuse.

**Data curation:** Masafumi Bekki.

**Formal analysis:** Ryuki Hashida.

**Investigation:** Takeshi Nago, Masafumi Bekki, Sohei Iwanaga, Eriko Higashi.

**Validation:** Hiroo Matsuse.

**Visualization:** Takeshi Nago.

**Writing – original draft:** Hiroshi Tajima.

**Writing – review & editing:** Hiroo Matsuse.

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
