## [Decision Letter · Decision Letter 0]

21 Jul 2021

PONE-D-21-17794

Electrically stimulated eccentric contraction during knee bending exercise in the dorsal position on a bed increases oxygen uptake

PLOS ONE

Dear Dr. Matsuse,

Thank you for submitting your manuscript to PLOS ONE. After careful consideration, we feel that it has merit but does not fully meet PLOS ONE’s publication criteria as it currently stands. Therefore, we invite you to submit a revised version of the manuscript that addresses the points raised during the review process.

We look forward to receiving your revised manuscript.

Kind regards,

Walid Kamal Abdelbasset, Ph.D.

Academic Editor

PLOS ONE

Journal Requirements:

2. Please amend your Methods section to state what type of informed consent was given (written, verbal, etc.).

Reviewers' comments:

Reviewer's Responses to Questions

**Comments to the Author**

1. Is the manuscript technically sound, and do the data support the conclusions?

Reviewer #1: Yes

Reviewer #2: Partly

Reviewer #3: Yes

2. Has the statistical analysis been performed appropriately and rigorously? 

Reviewer #1: Yes

Reviewer #2: I Don't Know

Reviewer #3: Yes

3. Have the authors made all data underlying the findings in their manuscript fully available?

Reviewer #1: Yes

Reviewer #2: Yes

Reviewer #3: Yes

4. Is the manuscript presented in an intelligible fashion and written in standard English?

Reviewer #1: Yes

Reviewer #2: Yes

Reviewer #3: Yes

5. Review Comments to the Author

Reviewer #1: Manuscript Number: PONE-D-21-17794

Overall, the idea of research is very interesting. However, there are some comments and suggestions.

Revision Suggestions

Title

I suggest some changes in the title as follow:

- Electrically stimulated eccentric contraction during non-weight bearing knee bending exercise increases oxygen uptake: a randomized controlled crossover trial

Abstract

The abstract is better designed as background (a brief one), purpose, methods (participants, study groups, outcome measures), results and conclusion.

Introduction

Line 82: please cite the references as (5-7)

Line 84: please cite the references as (8,9)

Materials and methods

- Ethical approval and trail registration are addressed.

- You need to clarify the study groups and the difference between them

- Line 105: correct (SD = 8.5)) to (SD = 8.5)

- Sample size was not calculated. Don’t you think the sample of 10 participants is relatively small and the results can't be generalized?

- The validity and reliability of assessment methods not stated

- For all figures, write the figure label under the corresponding figure. The label should not include statistical results. For example Fig.4. Comparing of changes of oxygen uptake.

- No need for fig 2 as figure 3 is sufficient

Statistical analysis: Satisfying

Results: Satisfying

Discussion: Satisfying

References

You need to update some references

References

Update some references is required

Reviewer #2: 1. Title need be revised and make it clear.

2. Can you write supine/prone? It will make easier to reader to read and matches with your manuscript.

3. Such as…Can you provide some examples? (line 72)

4. Would you write the full form of METS in 1st time (Metabolic…) (Line 77)

5. Can you provide operational definition? (line 84)

6. You need strong rational for study. (Line 91)

7. Why did you select healthy participants rather than clinical participants? (line 105)

8. Inclusion criteria are not clearly specified. Please write them in clear way. Study is in normal population but you have recommended for COVID, Space travelers ????? give strong points (109)

9. Can you explain why you choose this design of study? line 117

10. What is RISTA board and it’s importance in one sentence? line 120

11. Which techniques, need to be specified? line 126

12. Procedures are not clear.

13. Is it Rested or tested? line 133

14. Why in sitting position? Why not in supine? line. 134

15. If the testing was in sitting position than in which position did you performed test? Write it clearly. Intervention is not clear. line 137

16. Is this enough period for wash out? 139

17. You performed the test only one time? Or more than that? line 142

18. How did you measured the expired gas? line 149

19. Can you write about benefits of eccentric stimulation or contraction at introduction part? line 170

20. According to provided figure the test position is in supine. It will be clear for reader to write supine rather than dorsal lying. (figure 2, line 179)

21. Write supine. Use same word throughout the manuscript. line 187

22. Use same word through out the study rather then different. Don’t confuse readers. line 256

23. Correct the space. line 323

24. Is this the limitation of study ? line 328

25.

Reviewer #3: A randomized-controlled crossover clinical trial was conducted which aimed to assess oxygen uptake during deep knee bends but with and without a hybrid training system (HTS). Relative oxygen uptake was significantly higher during the HTS than without.

Minor revisions:

1- Line 201: Clarify if relative values of VO2, VCO2, and HR are differences between HTS and without. In this case a paired t-test was used for comparison as a post hoc test. Correct?

2- Line 198: Indicate the underlying covariance structure used in the linear mixed model and the criteria for selecting it.

3- The standard statistical terminology for “average” is “mean.”

4- Table 1: In addition to the frequencies, provide the percentage male.

6. PLOS authors have the option to publish the peer review history of their article (what does this mean?). If published, this will include your full peer review and any attached files.

Reviewer #1: No

Reviewer #2: **Yes: **1. Bishnu Dutta Acharya

2. Arpita Karki

Reviewer #3: No

---

## [Author Response · Author response to Decision Letter 0]

3 Sep 2021

PONE-D-21-17794

Electrically stimulated eccentric contraction during knee bending exercise in the dorsal position on a bed increases oxygen uptake 

Dear Dr. Matsuse,

Dear Editorial staff,

Thank you very much for your exceedingly kind comments for our study. We appreciate it very much. We answered all of your comments and revised the manuscript as follows. We submit the revised manuscript. Please contact me freely if you have any further questions.

Best regards,

Hiroo Matsuse

 

Journal Requirements:

1. Please ensure that your manuscript meets PLOS ONE's stylerequirements, including those for file naming. The PLOS ONE style templates can be found at

 and

2. Please amend your Methods section to state what type of informed consent was given (written, verbal, etc.).

A: We added the type of informed consent in the Methods section.

(Line 116)

A: We amended both of them according to the reviewer’s comments.

A: We deleted that from the acknowledge section and revised it.

5. Please include captions for your Supporting Information files at the end of your manuscript, and update any in-textcitations to match accordingly. Please see our Supporting Information guidelines for more information: http://journals.plos.org/plosone/s/supporting-information.

A: We added captions for figures and revised title and legend.

 

Reviewer #1: Manuscript Number: PONE-D-21-17794

Overall, the idea of research is very interesting. However, there are some comments and suggestions.

Revision Suggestions

Title

I suggest some changes in the title as follow:

- Electrically stimulated eccentric contraction during non-weight bearing knee bending exercise increases oxygen uptake: a randomized controlled crossover trial

A: We changed the title according to your comment. Moreover, we added the word “exploratory” in the title in response to your comment about the setting of sample size as follows: “Electrically stimulated eccentric contraction during non-weight bearing knee bending exercise in the supine position on a bed increases oxygen uptake: a randomized, controlled, exploratory crossover trial”

Abstract

The abstract is better designed as background (a brief one), purpose, methods (participants, study groups, outcome measures), results and conclusion.

A: Thank you for your comment. We added the words “in the supine position” in the abstract to response to reviewer #2’s comment. 

 (Line 46-47)

Introduction

Line 82: please cite the references as (5-7)

Line 84: please cite the references as (8,9)

A: Thank you for careful confirmation. We rewrote them as directed.

Materials and methods

- Ethical approval and trail registration are addressed.

A: We showed them on line 115-116.

- You need to clarify the study groups and the difference between them

A: We added the sentences as follows: “When participants were enrolled, the staff who was not involved in intervention and evaluation randomized the intervention sequence using a random number generator and sealed opaque envelopes with a block size of 5 to allocate to group A (the knee bending exercise test with HTS was performed firstly) or group B (the knee bending exercise test without HTS was performed firstly) (Fig 1). (Fig. 1).”. (Line 120-125 )

- Line 105: correct (SD = 8.5)) to(SD = 8.5)

A: We are sorry. We deleted the redundant sign “)”. (Line 118)

- Sample size was not calculated. Don’t you think the sample of 10

participants is relatively small and the results can't be generalized?

A: The sample of 10 participants may not be enough as you pointed out. Because there was not a preliminary study that evaluated the effect that HTS gave to oxygen uptake during knee bending exercise on a bed, we did not set a sample size. We wrote the sentence about this point on Line 226-227. Moreover, we added the expression of the exploratory study to the title to make it clear. 

Also, we added the sentences about the setting of the sample size to the session of the limitation in consultation with a statistician. (Line 357-361) 

We calculated the more common effect size (r) as a quantity of effect for an absolute number without changing by the sample size for a future study. (Line239-240) We calculated the effect size for t-test which we used to compare groups in this study again in the Result session. (Line262) However, we do not calculate the observed power for the results for an exploratory study. Furthermore, we added results of mean within subject difference (95% confidence interval) to reinforce the results of the primary outcome in this crossover trial. (Line260-262)

- The validity and reliability of assessment methods not stated

A: The validity and reliability analysis of VE and VO2 were performed as a comparison of intraclass correlation coefficients (ICCs) using Case 2 (2,1)

- For all figures, write the figure label under the corresponding figure.

 The label should not include statistical results. For example Fig.4.　Comparing of changes of oxygen uptake.

A: We revised and labeled them all as follows: 

Fig.3. Comparison of changes in oxygen uptake.

Fig.4. Comparison of changes in carbon dioxide production.

Fig.5. Comparison of changes in heart rate.

- No need for fig 2 as figure 3 is sufficient

A: We deleted Fig.2 and regulated the number of the figures.

Statistical analysis: Satisfying

Results: Satisfying

Discussion: Satisfying

A: Thank you for the confirmation.

 References

You need to update some references

References

Update some references is required

A: We confirmed and updated them using the EndNote.

Reviewer #2: 

1. Title need be revised and make it clear.

A: We changed the title according to reviewer 1’s advice. Moreover, we added the word “exploratory” in the title to solve reviewer 1’s comment about the setting of sample size and changed the words “dorsal” to “supine” according to your next comment as follows: “Electrically stimulated eccentric contraction during non-weight bearing knee bending exercise in the supine position on a bed increases oxygen uptake: a randomized, controlled, exploratory crossover trial”

2. Can you write supine/prone? It will make easier to reader to read and matches with your manuscript.

A: Thank you for your advice. We rewrote it using a word “supine” to make a spine position clear through the whole. (Line 2, 47, 150, 171, 182, 195, 283, 314, 346, 385, 415)

3. Such as…Can you provide some examples? (line 72)

A: We revised that sentence with examples as follows. “Bed rest causes similar disuse(3) although bed rest has been prescribed in the past for several other clinical conditions such as the rest for the intensive care or the quarantine for prevention of infection.”　（Line71-74）

4. Would you write the full form of METS in 1st time(Metabolic…) (Line

77)

A: Thanks for the input and we added a postscript to full spelling. Also, we deleted the full form from second METs.

5. Can you provide operational definition? (line 84)

A: We revised the sentence to include the operational definition as follows “the combined application of electrical stimulation (ES) and volitional contractions (VC) (performing ES during VC either non-simultaneously or simultaneously) is said to be more effective than ES or VC alone.” (Line 86-89) 

Moreover, we revised the next sentence from the ties with the above sentence as follows. “The hybrid training system (HTS) is one of the techniques that simultaneously combines VC and ES”. (Line89-90)

6. You need strong rational for study. (Line 91)

A: Many cases are considered as the clinical situation. However, we revised the sentence to include the explanation of the clinical situation as follows. “Knee bending exercise, one of the common exercises on a bed, is recommended for patients with disuse syndrome in a clinical situation such as the case that cannot walk for the reason of the motor disfunction or the case to be forced to lay for a remedial reason.” (Line99-102)

7. Why did you select healthy participants rather than clinical participants? (line 105)

A: This is an exploratory study. Moreover, the subjects of this study were healthy participants as preliminary research before intending for the clinical patients. As you know, we think that a pilot study like this becomes important before intending for the clinical patients. In future, we need the study in clinical patients in discussion. (Line369-370) 

We think that we have applied kinematical theory in the healthy subject to the clinical patients. We think that we can apply the possibility shown in healthy subjects in future in a larger field.

8. Inclusion criteria are not clearly specified. Please write them in clear way. Study is in normal population but you have recommended for COVID, Space travelers ????? give strong points (109)

A: We revised the inclusion criteria more definitely and rewrote it in detail as follows. “Inclusion criteria were applied: age between 20-50 years, nonsmokers, body mass index less than 30, doing normal physical activity, and normal physical function. A specialist in expert orthopedics and rehabilitation judged the normal physical function from the physical findings such as muscle strength, sensation, and range of motion according to the criteria of the Japanese Orthopedic Association.” (Line125-131)

Moreover, we revised that expression as follows. “These findings showed the possibility that we might strengthen the metabolic stress from increased physical activity by combining HTS with a knee bending exercise that is easily performed by bed ridden patients” (Line352-355)

9. Can you explain why you choose this design of study? line 117

A: The first reason is advice from a statistician of our institution. Generally, we know that this crossover design is suitable for the disease that a case is hard to recruit. However, we think that it is useful to apply this design to the intervention study using the electrical stimulation like this study because we are very difficult to perform this intervention with the complete blindness (both researcher and subject) unlike a study using the placebo medicine. Subjects are on their own controls. The within-patient variation is less than between-patient variation. Therefore, we think that we can evaluate the intervention effect with minimizing the influence of the individual difference by comparing the presence or absence of electrically stimulated muscle contraction during volitional exercise in the same individuals.

We added a postscript to that reason as follows. (Line141-144) “It is very difficult for both subject and researcher to completely blind presence or absence of electrical stimulation. Moreover, we need to minimize the influence of the individual difference. Therefore, we chose a crossover trial in this study.”

10. What is RISTA board and it’s importance in one sentence? line 120

A: RISTA board is a device to facilitate knee bending exercises in the supine position on a bed. We revised the sentence more clearly and added a postscript to the explanation of the device as follows. “We measured gas exchange while subjects did knee bending exercises with a RISTA board (NIPPON SIGMAX Co, Tokyo, Japan) at the lowest exercise resistance load using the following protocol after a rest for 30 minutes. The RISTA board is a Japanese medical device to facilitate knee bending exercises in the supine position on a bed.” (Line145-150)

11. Which techniques, need to be specified? line 126

A: We added a postscript to the technique of the system in detail as follows. “The AE-100i consists of a microcomputer, a hot wire flowmeter, and a gas analyzer, which contains a sampling tube, filter, suction pump, oxygen (O2) analyzer made by a paramagnetic O2 transducer, and an infrared carbon dioxide (CO2) analyzer. Ventilatory parameters were measured using a hot-wire flow meter, and the flow meter was calibrated with a syringe of known volume (2.0l). A zirconium sensor and an infrared absorption analyzer, respectively, measured O2 and CO2 concentrations. The gas analyzer was calibrated to known standard gas levels (O2 15.16%, CO2 5.023%) before each test.” (Line154-162 )

12. Procedures are not clear.

A: We already described the method of the exhalation gas analysis. (Line151-166 ) Moreover, we added the next sentences and revised the first paragraph of the Intervention to make the procedures clearer as follows. “Participants performed the knee bending exercise tests with or without HTS according to a sequence assigned before the exercise test. Next, participants transitioned to the alternative exercise test on different days separated by an interval of one week as a wash out period. Therefore, the participant who first performed the exercise test with HTS performed the exercise test without HTS next.” (Line174-179). 

Also, we revised the next sentence to make the exercise tests clear as follows. “In the interventional exercise, subjects performed the knee bending exercise with reciprocal 2-second (45degree/sec) knee flexion and extension contractions at the speed of 1Hz using a metronome for 5 minutes with HTS or without HTS in the supine position on a flat surface after warming up at a slow speed for 3 minutes.” (Line179-183) 

At the same time, we revised the first paragraph of Subjetcs in Method session. 

13. Is it Rested or tested? line 133

A: That is “rested”. (Line170) We revised the sentence associated with other opinions.

14. Why in sitting position? Why not in supine? line. 134

A: We revised the sentence as follows to make the purpose clear. “At first subjects rested for five minutes in a sitting position to evaluate of the basic inactivity quiet metabolic cost while sitting quietly”(Line172-173)

It is not our purpose to make the participant rest quietly in bed. We set it as an evaluation of the basic inactivite metabolic cost as 1 MET (sitting quietly). Lying quietly is 1 MET as an inactivite cost also as you mentioned. The Ministry of Health, Labour and Welfare in Japan defines sitting quietly as 1MET; the unit of physical activity or motion. sitting quietly is the first step of a physical activity to achieve an increase from decubitus in a rehabilitation clinic. Therefore, we can estimate the clinical benefit of an intervention by knowing the metabolic cost during the intervention as well as for sitting quietly. Also, we think that we can confirm that intervention in this study is definitely the physical activity from the exercise by comparing it with the conditions measured during the resting state. Because of this, we showed the results of the resting state in figures. 

Moreover, we added the sentence to distinguish the “rest” before the test definitely as follows. “Participants rested for 30 min in the supine position to minimize the influence of physical activity before the exercise test..” (Line170-172)

15. If the testing was in sitting position than in which position did you performed test? Write it clearly. Intervention is not clear. line137

A: Like the figure 2, we performed the exercise testing (intervention) in the supine position on a bed. In response to your former comment, we added intervention contents to the other places of the paragraph clearly and made modifications as follows. “In the interventional exercise, subjects performed the knee bending exercise with reciprocal 2-second (45degree/sec) knee flexion and extension contractions at the speed of 1Hz using a metronome for 5 minutes with HTS or without HTS in the supine position on a flat surface after warming up at a slow speed for 3 minutes (Fig.2).” (Line179-183)

16. Is this enough period for wash out? 139

A: We think that that was enough. For example, a study which evaluated muscle fatigue from exercise adopts one week as a wash out. (PMID: 33232629、PMID: 25494054) Also, there were only two days as a wash out in a study that evaluated metabolism after aerobic exercise (2km running). (PMID: 28622349) Moreover, the influence of muscle fatigue seems to be restored largely within 3 days. (PMID: 24435468) 

Besides, the exercise intensity that we performed in this study is considerably low intensity, equivalent to light activities of daily living. However, there may be muscle damage due to the electrical eccentric contraction although there adverse events including muscle soreness in this study. Also, in another study, we have showed that there was little to no muscle damage after HTS in the arm. (PMID: 17317931)

Moreover, the statistical method we used provides a method to test carryover effects. Carryover effects will cause the difference between the two treatments (intervention) to be different in the two time periods. When the interaction is significant, it indicates the presence of carryover. If the carryover effect is not significantly between two sequences. There were no significant differences between sequences in this study.

Therefore, we judged that the influence of the exercise intervention in this study did not have any carry over. However, we added a postscript about this matter in the limitation section in Discussion because we might not be able to completely deny it as follows. “Furthermore, the group effect (sequence) was absent, but it is unclear whether we could wash out the influence of HTS in one week. Therefore, a study not implementing a crossover trial may be necessary.” (Line 370-373)

17. You performed the test only one time? Or more than that? line 142

A: We tested once for each intervention (with HTS or without HTS) on different days, respectively, as we wrote in the first half of Intervention section and figure 1. We rewrote it for every test (with HTS or without HTS) to be plain as follows. “Successively, subjects performed 2 minutes cool down. Therefore, they had the total exercise time of 10 minutes in each of two tests in consideration of muscle fatigue” (Line184-186)

18. How did you measured the expired gas? line 149

A: We showed how to measure the expired gas at Line 145-166. 

Because it is the measurement that is commonly used widely for clinical medicine as well as clinical studies , we think that it is not so special s hat it is necessary to describe the detailed technique in the main text.

 Because we used the automated breath by breath system, we can obtain data (VO2, VCO2, and VE) of the gas for each breath. For analysis, we averaged the data of each subject, respectively, (respiratory frequency is different for every individual) obtained in the last 2 minutes of the exercise time of 5 minutes. We revised the method that we averaged a little in detail as follows. “Therefore, we averaged the data of HR and expired gas during the last 2 minutes of the interventional exercise time of 5 minutes of each test and used them for data analysis.” (Line190-192)

19. Can you write about benefits of eccentric stimulation or contraction at introduction part? line 170

A: We added that benefit in Introduction part with new references as follows. 

“HTS utilizes electrically stimulated eccentric contractions. Eccentric contractions are important for muscle strengthening(1). Furthermore, electrically stimulated eccentric contractions are recognized as a muscle strengthening technique, because we can obtain bigger muscle torque (exercise load) than with concentric contractions alone(2, 3). It has been reported that HTS can increase both muscle strength and mass even when exercise intensity is relatively low(4-7).” (Line91-98)

1. Dudley GA, Tesch PA, Miller BJ, Buchanan P. Importance of eccentric actions in performance adaptations to resistance training. Aviat Space Environ Med. 1991;62(6):543-50.

2. Seger JY, Thorstensson A. Electrically evoked eccentric and concentric torque-velocity relationships in human knee extensor muscles. Acta Physiol Scand. 2000;169(1):63-9.

3. Pain MT, Young F, Kim J, Forrester SE. The torque-velocity relationship in large human muscles: maximum voluntary versus electrically stimulated behaviour. J Biomech. 2013;46(4):645-50.

4. Matsuse H, Shiba N, Umezu Y, Nago T, Tagawa Y, Kakuma T, et al. Muscle training by means of combined electrical stimulation and volitional contraction. Aviat Space Environ Med. 2006;77(6):581-5.

5. Iwasaki T, Shiba N, Matsuse H, Nago T, Umezu Y, Tagawa Y, et al. Improvement in knee extension strength through training by means of combined electrical stimulation and voluntary muscle contraction. Tohoku J Exp Med. 2006;209(1):33-40.

6. Takano Y, Haneda Y, Maeda T, Sakai Y, Matsuse H, Kawaguchi T, et al. Increasing muscle strength and mass of thigh in elderly people with the hybrid-training method of electrical stimulation and volitional contraction. Tohoku J Exp Med. 2010;221(1):77-85.

7. Rabe KG, Matsuse H, Jackson A, Segal NA. Evaluation of the Combined Application of Neuromuscular Electrical Stimulation and Volitional Contractions on Thigh Muscle Strength, Knee Pain, and Physical Performance in Women at Risk for Knee Osteoarthritis: A Randomized Controlled Trial. PM R. 2018;10(12):1301-10.

20. According to provided figure the test position is in supine. It will be clear for reader to write supine rather than dorsal lying. (figure 2,line 179)

A: We revised the expression in "supine position" at Title to make the test position clear. Also, according to the opinion of the other reviewer, we deleted figure 2 and revised the explanation.

21. Write supine. Use same word throughout the manuscript. line 187

A: Associated with the comment mentioned above, we revised all of it.

22. Use same word through out the study rather then different. Don’t confuse readers. line 256

A: Associated with the comment mentioned above, we rewrote it to “in the supine position”. (Line 2, 47, 150, 171, 182, 195, 283, 314, 346, 385, 415)

23. Correct the space. line 323

A: We corrected it.

24. Is this the limitation of study ? line 328

A: We deleted that sentence.

 

Reviewer #3: A randomized-controlled crossover clinical trial was conducted which aimed to assess oxygen uptake during deep knee bends but with and without a hybrid training system (HTS). Relative oxygen uptake was significantly higher during the HTS than without.

Minor revisions:

1-Line 201: Clarify if relative values of VO2, VCO2, and HR are differences between HTS and without. In this case a paired t-test was used for comparison as a post hoc test. Correct?

A: Because we had paired quantitative measurements from two groups. We corrected it to paired Student’s t-test. (Line237)

2- Line 198: Indicate the underlying covariance structure used in the linear mixed model and the criteria for selecting it.

A: We set subjectID(sequence) as random effects covariance parameters for random effects model like a typical model for crossover trials. We rewrote it in detail. (Line234-236)

3- The standard statistical terminology for “average” is “mean.”

A: We changed that word to “mean” at Line263.

4- Table 1: In addition to the frequencies, provide the percentage male.

A: We added the percentage male.

 

----

6. PLOS authors have the option to publish the peer review history of their article (what does this mean?). If published, this will include your full peer review and any attached files. 

Do you want your identity to be public for this peer review? For information about this choice, including consent withdrawal, please see our Privacy Policy.

Reviewer 

#1: No

Reviewer #2: Yes: 1. Bishnu Dutta Acharya

2. Arpita Karki

Reviewer #3: No

While revising your submission, please upload your figure files to the Preflight Analysis and Conversion Engine (PACE) digital diagnostic tool,

https://pacev2.apexcovantage.com/. PACE helpsensure that figures meet

PLOS requirements. To use PACE, you must first register as a user.

Registration is free. Then, login and navigate to the UPLOAD tab, where

you will find detailed instructions on how to use the tool. If you encounter any issues or have any questions when using PACE, please email PLOS at figures@plos.org. Please note that Supporting Information files

do not need this step.

In compliance with data protection regulations, you may request that we

remove your personal registration details at any time. (Remove my

information/details). Please contact the publication office if you have

any questions.

---

## [Decision Letter · Decision Letter 1]

13 Sep 2021

PONE-D-21-17794R1Electrically stimulated eccentric contraction during non-weight bearing knee bending exercise in the supine position on a bed increases oxygen uptake: a randomized, controlled, exploratory crossover trialPLOS ONE

Dear Dr. Matsuse,

Thank you for submitting your manuscript to PLOS ONE. After careful consideration, we feel that it has merit but does not fully meet PLOS ONE’s publication criteria as it currently stands. Therefore, we invite you to submit a revised version of the manuscript that addresses the points raised during the review process. Please submit your revised manuscript by Oct 28 2021 11:59PM. If you will need more time than this to complete your revisions, please reply to this message or contact the journal office at plosone@plos.org. Please include the following items when submitting your revised manuscript:A rebuttal letter that responds to each point raised by the academic editor and reviewer(s). You should upload this letter as a separate file labeled 'Response to Reviewers'.A marked-up copy of your manuscript that highlights changes made to the original version. You should upload this as a separate file labeled 'Revised Manuscript with Track Changes'.An unmarked version of your revised paper without tracked changes. You should upload this as a separate file labeled 'Manuscript'.If applicable, we recommend that you deposit your laboratory protocols in protocols.io to enhance the reproducibility of your results. Protocols.io assigns your protocol its own identifier (DOI) so that it can be cited independently in the future. For instructions see: https://journals.plos.org/plosone/s/submission-guidelines#loc-laboratory-protocols. Additionally, PLOS ONE offers an option for publishing peer-reviewed Lab Protocol articles, which describe protocols hosted on protocols.io. Read more information on sharing protocols at https://plos.org/protocols?utm_medium=editorial-email&utm_source=authorletters&utm_campaign=protocols.

We look forward to receiving your revised manuscript.

Kind regards,

Walid Kamal Abdelbasset, Ph.D.

Academic Editor

PLOS ONE

Journal Requirements:

Additional Editor Comments (if provided):

Reviewers' comments:

Reviewer's Responses to Questions

**Comments to the Author**

1. If the authors have adequately addressed your comments raised in a previous round of review and you feel that this manuscript is now acceptable for publication, you may indicate that here to bypass the “Comments to the Author” section, enter your conflict of interest statement in the “Confidential to Editor” section, and submit your "Accept" recommendation.

Reviewer #1: All comments have been addressed

Reviewer #2: All comments have been addressed

Reviewer #3: (No Response)

2. Is the manuscript technically sound, and do the data support the conclusions?

Reviewer #1: Yes

Reviewer #2: Yes

Reviewer #3: Yes

3. Has the statistical analysis been performed appropriately and rigorously? 

Reviewer #1: Yes

Reviewer #2: Yes

Reviewer #3: Yes

4. Have the authors made all data underlying the findings in their manuscript fully available?

Reviewer #1: Yes

Reviewer #2: Yes

Reviewer #3: Yes

5. Is the manuscript presented in an intelligible fashion and written in standard English?

Reviewer #1: Yes

Reviewer #2: Yes

Reviewer #3: Yes

6. Review Comments to the Author

Reviewer #1: Thanks to the authors for their detailed response and corrections. No further changes are suggested. the authors have done all required modifications.

Reviewer #2: 1. Still the title is not in the PICO Format. Is it important to write "on a bed" try to shorten the title.

2. short tile doesn't match the main title.

3. reference for the line 100-102

4. line 2017: Add reference to support this point.

Reviewer #3: Minor revisions:

1. The following prior comment was not adequately addressed. Indicate the underlying covariance structure used in the linear mixed model and the criteria for selecting it.

The top 3 most common covariance structures are compound symmetry, unstructured, and autoregressive.

2- Table 1: Include the corresponding percentage male.

7. PLOS authors have the option to publish the peer review history of their article (what does this mean?). If published, this will include your full peer review and any attached files.

Reviewer #1: No

Reviewer #2: **Yes: **Bishnu Dutta Acharya

Reviewer #3: No

---

## [Author Response · Author response to Decision Letter 1]

13 Oct 2021

Dear Editorial staff,

Thank you very much for your exceedingly kind comments and advice for our study. We appreciate it very much. We answered all of your comments and revised the manuscript as follows. We submit the revised manuscript. Please contact me freely if you have any further questions.

Best regards,

Hiroo Matsuse

A: We apologize that we were not able to confirm the reference list enough. We think the retracted paper was not included. 

Because the paper of Omoto et al. (19) was not published in the pubmed, we have got a wrong reproduction. Also, the notation of his name was revised from the beginning of publication from Ohmoto to Omoto. We revised it in the latest information. 

Because my paper (18) was a Web version, we updated it about the information (doi). Similarly, we confirmed the pubmed one by one and added a postscript to information of doi or updated the information. Also, we revised Epub date because the Epub date of the downloaded nbib file from pubmed was different from the PubMed site. We confirmed them one by one in the website of pubmed and revised them which have the information of Epub in the website of pubmed.

Because "PMC" was repeated in much PMCID, we deleted them. However, PMCID and doi were not published when we confirmed a latest PLOS ONE’s paper. Should we delete them? At least PMCID seemed to be unnecessary.

We download an endnote style (https://endnote.com/style_download/plos-public-library-of-science-all-journals/ ) from Submission Guidelines (https://journals.plos.org/plosone/s/submission-guidelines#loc-references) and used it, and, however, we think whether you are different from the “Vancouver” style a little. If there are unnecessary information and necessary information, please tell us.

Moreover, we added three references in this revised manuscript (20, 22, 33). Therefore, the reference numbers were changed. 

We showed revised or changed points by a deficit as follows.

References

1. Nagaraja MP, Jo H. The Role of Mechanical Stimulation in Recovery of Bone Loss-High versus Low Magnitude and Frequency of Force. Life (Basel). 2014;4(2):117-30. Epub 2014/11/06. doi: 10.3390/life4020117. PubMed PMID: 25370188; PubMed Central PMCID: PMC4187165.

2. Valenzuela PL, Morales JS, Pareja-Galeano H, Izquierdo M, Emanuele E, de la Villa P, et al. Physical strategies to prevent disuse-induced functional decline in the elderly. Ageing Res Rev. 2018;47:80-8. Epub 2018/07/18. doi: 10.1016/j.arr.2018.07.003. PubMed PMID: 30031068.

3. Bloomfield SA. Changes in musculoskeletal structure and function with prolonged bed rest. Med Sci Sports Exerc. 1997;29(2):197-206. Epub 1997/02/01. doi: 10.1097/00005768-199702000-00006. PubMed PMID: 9044223.

4. Voet NB, van der Kooi EL, van Engelen BG, Geurts AC. Strength training and aerobic exercise training for muscle disease. Cochrane Database Syst Rev. 2019;12(12):CD003907. Epub 2019/12/07. doi: 10.1002/14651858.CD003907.pub5. PubMed PMID: 31808555; PubMed Central PMCID: PMC6953420.

5. Maffiuletti NA, Green DA, Vaz MA, Dirks ML. Neuromuscular Electrical Stimulation as a Potential Countermeasure for Skeletal Muscle Atrophy and Weakness During Human Spaceflight. Front Physiol. 2019;10:1031. Epub 2019/08/29. doi: 10.3389/fphys.2019.01031. PubMed PMID: 31456697; PubMed Central PMCID: PMC6700209.

6. Brower RG. Consequences of bed rest. Crit Care Med. 2009;37(10 Suppl):S422-8. Epub 2010/02/06. doi: 10.1097/CCM.0b013e3181b6e30a. PubMed PMID: 20046130.

7. Dirks ML, Wall BT, van Loon LJC. Interventional strategies to combat muscle disuse atrophy in humans: focus on neuromuscular electrical stimulation and dietary protein. J Appl Physiol (1985). 2018;125(3):850-61. Epub 2017/9/28. doi: 10.1152/japplphysiol.00985.2016. PubMed PMID: 28970205.

8. Gomes Neto M, Oliveira FA, Reis HF, de Sousa Rodrigues E, Jr., Bittencourt HS, Oliveira Carvalho V. Effects of Neuromuscular Electrical Stimulation on Physiologic and Functional Measurements in Patients With Heart Failure: A SYSTEMATIC REVIEW WITH META-ANALYSIS. J Cardiopulm Rehabil Prev. 2016;36(3):157-66. Epub 2016/01/20. doi: 10.1097/HCR.0000000000000151. PubMed PMID: 26784735.

9. Hamada T, Hayashi T, Kimura T, Nakao K, Moritani T. Electrical stimulation of human lower extremities enhances energy consumption, carbohydrate oxidation, and whole body glucose uptake. J Appl Physiol (1985). 2004;96(3):911-6. Epub 2003/10/31. doi: 10.1152/japplphysiol.00664.2003. PubMed PMID: 14594864.

10. Paillard T. Combined application of neuromuscular electrical stimulation and voluntary muscular contractions. Sports Med. 2008;38(2):161-77. doi: 10.2165/00007256-200838020-00005. PubMed PMID: 18201117.

11. Dudley GA, Tesch PA, Miller BJ, Buchanan P. Importance of eccentric actions in performance adaptations to resistance training. Aviat Space Environ Med. 1991;62(6):543-50. PubMed PMID: 1859341.

12. Seger JY, Thorstensson A. Electrically evoked eccentric and concentric torque-velocity relationships in human knee extensor muscles. Acta Physiol Scand. 2000;169(1):63-9. Epub 2000/04/12. doi: 10.1046/j.1365-201x.2000.00694.x. PubMed PMID: 10759612.

13. Pain MT, Young F, Kim J, Forrester SE. The torque-velocity relationship in large human muscles: maximum voluntary versus electrically stimulated behaviour. J Biomech. 2013;46(4):645-50. Epub 2013/01/11. doi: 10.1016/j.jbiomech.2012.11.052. PubMed PMID: 23313275.

14. Matsuse H, Shiba N, Umezu Y, Nago T, Tagawa Y, Kakuma T, et al. Muscle training by means of combined electrical stimulation and volitional contraction. Aviat Space Environ Med. 2006;77(6):581-5. PubMed PMID: 16780234.

15. Iwasaki T, Shiba N, Matsuse H, Nago T, Umezu Y, Tagawa Y, et al. Improvement in knee extension strength through training by means of combined electrical stimulation and voluntary muscle contraction. Tohoku J Exp Med. 2006;209(1):33-40. doi: 10.1620/tjem.209.33. PubMed PMID: 16636520.

16. Takano Y, Haneda Y, Maeda T, Sakai Y, Matsuse H, Kawaguchi T, et al. Increasing muscle strength and mass of thigh in elderly people with the hybrid-training method of electrical stimulation and volitional contraction. Tohoku J Exp Med. 2010;221(1):77-85. doi: 10.1620/tjem.221.77. PubMed PMID: 20453461.

17. Rabe KG, Matsuse H, Jackson A, Segal NA. Evaluation of the Combined Application of Neuromuscular Electrical Stimulation and Volitional Contractions on Thigh Muscle Strength, Knee Pain, and Physical Performance in Women at Risk for Knee Osteoarthritis: A Randomized Controlled Trial. PM R. 2018;10(12):1301-10. Epub 2018/05/29. doi: 10.1016/j.pmrj.2018.05.014. PubMed PMID: 29852286; PubMed Central PMCID: PMC6719317.

18. Matsuse H, Shiba N, Takano Y, Yamada S, Ohshima H, Tagawa Y. Cycling exercise to resist electrically stimulated antagonist increases oxygen uptake in males: pilot study. J Rehabil Res Dev. 2013;50(4):545-54. doi: 10.1682/jrrd.2012.04.0067. PubMed PMID: 23934874.

19. Omoto M, Matsuse H, Takano Y, Yamada S, Ohshima H, Tagawa Y, et al. Oxygen Uptake during Aerobic Cycling Exercise Simultaneously Combined with Neuromuscular Electrical Stimulation of Antagonists. J Nov Physiother. 2013;3:185. Epub 2013/11/07. doi: 10.4172/2165-7025.1000185.

20. Cameron S, Ball I, Cepinskas G, Choong K, Doherty TJ, Ellis CG, et al. Early mobilization in the critical care unit: A review of adult and pediatric literature. J Crit Care. 2015;30(4):664-72. Epub 2015/04/08. doi: 10.1016/j.jcrc.2015.03.032. PubMed PMID: 25987293.

21. Gaesser GA, Poole DC. The slow component of oxygen uptake kinetics in humans. Exerc Sport Sci Rev. 1996;24:35-71. Epub 1996/01/01. PubMed PMID: 8744246.

22. Ward AR, Shkuratova N. Russian electrical stimulation: the early experiments. Phys Ther. 2002;82(10):1019-30. Epub 2002/09/28. PubMed PMID: 12350217.

23. Matsuse H, Segal NA, Rabe KG, Shiba N. The Effect of Neuromuscular Electrical Stimulation During Walking on Muscle Strength and Knee Pain in Obese Women With Knee Pain: A Randomized Controlled Trial. Am J Phys Med Rehabil. 2020;99(1):56-64. Epub 2019/10/09. doi: 10.1097/PHM.0000000000001319. PubMed PMID: 31592880.

24. Tsukada Y, Matsuse H, Shinozaki N, Takano Y, Nago T, Shiba N. Combined Application of Electrically Stimulated Antagonist Muscle Contraction and Volitional Muscle Contraction Prevents Muscle Strength Weakness and Promotes Physical Function Recovery After Total Knee Arthroplasty: A Randomized Controlled Trial. Kurume Med J. 2020;65(4):145-54. Epub 2019/11/13. doi: 10.2739/kurumemedj.MS654007. PubMed PMID: 31723080.

25. Oates BR, Glover EI, West DW, Fry JL, Tarnopolsky MA, Phillips SM. Low-volume resistance exercise attenuates the decline in strength and muscle mass associated with immobilization. Muscle Nerve. 2010;42(4):539-46. Epub 2010/07/27. doi: 10.1002/mus.21721. PubMed PMID: 20658567.

26. Alkner BA, Tesch PA. Efficacy of a gravity-independent resistance exercise device as a countermeasure to muscle atrophy during 29-day bed rest. Acta Physiol Scand. 2004;181(3):345-57. Epub 2004/06/16. doi: 10.1111/j.1365-201X.2004.01293.x. PubMed PMID: 15196095.

27. Berg HE, Tesch A. A gravity-independent ergometer to be used for resistance training in space. Aviat Space Environ Med. 1994;65(8):752-6. Epub 1994/08/01. PubMed PMID: 7980338.

28. Burke D, Gorman E, Stokes D, Lennon O. An evaluation of neuromuscular electrical stimulation in critical care using the ICF framework: a systematic review and meta-analysis. Clin Respir J. 2016;10(4):407-20. Epub 2014/11/26. doi: 10.1111/crj.12234. PubMed PMID: 25353646.

29. Iwanaga S, Hashida R, Takano Y, Bekki M, Nakano D, Omoto M, et al. Hybrid Training System Improves Insulin Resistance in Patients with Nonalcoholic Fatty Liver Disease: A Randomized Controlled Pilot Study. Tohoku J Exp Med. 2020;252(1):23-32. Epub 2020/08/31. doi: 10.1620/tjem.252.23. PubMed PMID: 32863329.

30. Numata H, Nakase J, Inaki A, Mochizuki T, Oshima T, Takata Y, et al. Effects of the belt electrode skeletal muscle electrical stimulation system on lower extremity skeletal muscle activity: Evaluation using positron emission tomography. J Orthop Sci. 2016;21(1):53-6. Epub 2015/11/21. doi: 10.1016/j.jos.2015.09.003. PubMed PMID: 26755387.

31. Forestieri P, Bolzan DW, Santos VB, Moreira RSL, de Almeida DR, Trimer R, et al. Neuromuscular electrical stimulation improves exercise tolerance in patients with advanced heart failure on continuous intravenous inotropic support use-randomized controlled trial. Clin Rehabil. 2018;32(1):66-74. Epub 2017/06/21. doi: 10.1177/0269215517715762. PubMed PMID: 28633534.

32. Dalise S, Cavalli L, Ghuman H, Wahlberg B, Gerwig M, Chisari C, et al. Biological effects of dosing aerobic exercise and neuromuscular electrical stimulation in rats. Sci Rep. 2017;7(1):10830. Epub 2017/09/09. doi: 10.1038/s41598-017-11260-7. PubMed PMID: 28883534; PubMed Central PMCID: PMC5589775.

33. Iwatsu K, Yamada S, Iida Y, Sampei H, Kobayashi K, Kainuma M, et al. Feasibility of neuromuscular electrical stimulation immediately after cardiovascular surgery. Arch Phys Med Rehabil. 2015;96(1):63-8. Epub 2014/09/09. doi: 10.1016/j.apmr.2014.08.012. PubMed PMID: 25218214.

34. Fontes Cerqueira TC, Cerqueira Neto ML, Cacau LAP, Oliveira GU, Silva Junior WMD, Carvalho VO, et al. Ambulation capacity and functional outcome in patients undergoing neuromuscular electrical stimulation after cardiac valve surgery: A randomised clinical trial. Medicine (Baltimore). 2018;97(46):e13012. Epub 2018/11/16. doi: 10.1097/MD.0000000000013012. PubMed PMID: 30431575; PubMed Central PMCID: PMC6257613.

35. Hainaut K, Duchateau J. Neuromuscular electrical stimulation and voluntary exercise. Sports Med. 1992;14(2):100-13. Epub 1992/08/01. doi: 10.2165/00007256-199214020-00003. PubMed PMID: 1509225.

36. Theurel J, Lepers R, Pardon L, Maffiuletti NA. Differences in cardiorespiratory and neuromuscular responses between voluntary and stimulated contractions of the quadriceps femoris muscle. Respir Physiol Neurobiol. 2007;157(2-3):341-7. Epub 2006/12/15. doi: 10.1016/j.resp.2006.12.002. PubMed PMID: 17210271.

37. Liu M, Luo J, Zhou J, Zhu X. Intervention effect of neuromuscular electrical stimulation on ICU acquired weakness: A meta-analysis. Int J Nurs Sci. 2020;7(2):228-37. Epub 2020/03/10. doi: 10.1016/j.ijnss.2020.03.002. PubMed PMID: 32685621; PubMed Central PMCID: PMC7355203.

38. Nakamura K, Nakano H, Naraba H, Mochizuki M, Hashimoto H. Early rehabilitation with dedicated use of belt-type electrical muscle stimulation for severe COVID-19 patients. Crit Care. 2020;24(1):342. Epub 2020/06/15. doi: 10.1186/s13054-020-03080-5. PubMed PMID: 32539827; PubMed Central PMCID: PMC7294763.

39. Swain DP, Leutholtz BC. Heart rate reserve is equivalent to %VO2 reserve, not to %VO2max. Med Sci Sports Exerc. 1997;29(3):410-4. Epub 1997/03/01. doi: 10.1097/00005768-199703000-00018. PubMed PMID: 9139182.

Additional Editor Comments (if provided):

Q: Is it safe to use fNMES for cardiac pateints?

A: It is safe if we use it appropriately. Of course, it is necessary to avoid the use to pacemaker wearers. However, we may not have any problem depending on how to use (Egger F, 2019). A useful of the electrical stimulation therapy in patients with chronic heart failure is shown in a systematic review (Gomes Neto M, 2016). Functional electrical stimulation of lower limb muscles is considered as an alternative mode of exercise training in chronic heart failure (Parissis J, 2015). Moreover, it has been shown that the use of NMES is safe in patients immediately after cardiovascular surgery (Iwatsu K, 2015). 

 We rewrote the sentence about the safety with a reference as follows.

“It has been shown that the use of NMES is safe in patients after cardiovascular surgery (Iwatsu K, 2015), but NMES did not improve functional capacity in cardiac valve surgery patients in the immediate postoperative period.” 

References

Egger F, Hofer C, Hammerle FP, Lofler S, Nurnberg M, Fiedler L, et al. Influence of electrical stimulation therapy on permanent pacemaker function. Wien Klin Wochenschr. 2019;131(13-14):313-20. Epub 2019/04/27. doi: 10.1007/s00508-019-1494-5. PubMed PMID: 31025164.

Gomes Neto M, Oliveira FA, Reis HF, de Sousa Rodrigues E, Jr., Bittencourt HS, Oliveira Carvalho V. Effects of Neuromuscular Electrical Stimulation on Physiologic and Functional Measurements in Patients With Heart Failure: A SYSTEMATIC REVIEW WITH META-ANALYSIS. J Cardiopulm Rehabil Prev. 2016;36(3):157-66. Epub 2016/01/20. doi: 10.1097/HCR.0000000000000151. PubMed PMID: 26784735.

Parissis J, Farmakis D, Karavidas A, Arapi S, Filippatos G, Lekakis J. Functional electrical stimulation of lower limb muscles as an alternative mode of exercise training in chronic heart failure: practical considerations and proposed algorithm. Eur J Heart Fail. 2015;17(12):1228-30. Epub 2015/10/16. doi: 10.1002/ejhf.409. PubMed PMID: 26466970.

Iwatsu K, Yamada S, Iida Y, Sampei H, Kobayashi K, Kainuma M, et al. Feasibility of neuromuscular electrical stimulation immediately after cardiovascular surgery. Arch Phys Med Rehabil. 2015;96(1):63-8. Epub 2014/09/09. doi: 10.1016/j.apmr.2014.08.012. PubMed PMID: 25218214.

Reviewers' comments:

Reviewer's Responses to Questions

Comments to the Author

1. If the authors have adequately addressed your comments raised in a previous round of review and you feel that this manuscript is now acceptable for publication, you may indicate that here to bypass the “Comments to the Author” section, enter your conflict of interest statement in the “Confidential to Editor” section, and submit your "Accept" recommendation.

Reviewer #1: All comments have been addressed

Reviewer #2: All comments have been addressed

Reviewer #3: (No Response)

2. Is the manuscript technically sound, and do the data support the conclusions?

Reviewer #1: Yes

Reviewer #2: Yes

Reviewer #3: Yes

3. Has the statistical analysis been performed appropriately and rigorously?

Reviewer #1: Yes

Reviewer #2: Yes

Reviewer #3: Yes

4. Have the authors made all data underlying the findings in their manuscript fully available?

Reviewer #1: Yes

Reviewer #2: Yes

Reviewer #3: Yes

5. Is the manuscript presented in an intelligible fashion and written in standard English?

Reviewer #1: Yes

Reviewer #2: Yes

Reviewer #3: Yes

6. Review Comments to the Author

Reviewer #1: Thanks to the authors for their detailed response and corrections. No further changes are suggested. the authors have done all required modifications.

A: Thank you for your polite review.

Reviewer #2: 

1. Still the title is not in the PICO Format. Is it important to write "on a bed" try to shorten the title.

A: We deleted “on a bed".

　　“Electrically stimulated eccentric contraction during non-weight bearing knee bending exercise in the supine position increases oxygen uptake: a randomized, controlled, exploratory crossover trial”

2. short tile doesn't match the main title.

A: We changed the short title to “Oxygen uptake during knee bending exercise combined with electrical stimulation” to state the topic of the study. 

(70 characters)

3. reference for the line 100-102

A: We think that the lower extremity bending exercise is not special, and it is the general physical therapy as an early mobilization which was recommended for critically ill patients. We revised the sentence as one of an early mobilization. We revised the sentence and added the reference (Cameron S, 2015) which shows the recommendation of the early mobilization for critically ill patients.

“Knee bending exercise, one of the common early mobilizations on a bed, is recommended for critically ill patients [20].”

Reference

Cameron S, Ball I, Cepinskas G, Choong K, Doherty TJ, Ellis CG, et al. Early mobilization in the critical care unit: A review of adult and pediatric literature. J Crit Care. 2015;30(4):664-72. Epub 2015/04/08. doi: 10.1016/j.jcrc.2015.03.032. PubMed PMID: 25987293.

4. line 2017: Add reference to support this point.

A: We added the reference about the standard Russian waveform (Ward AR, 2002). Moreover, we showed our past studies using this parameter.

“ES parameters were based on a standard Russian waveform (Ward AR, 2002) in which a 5,000 Hz carrier frequency is modulated at 40 Hz (2.4 ms on, 22.6 ms off) to deliver a rectangular voltage biphasic pulse [16][17-19, 23, 24].”

Reference

Ward AR, Shkuratova N. Russian electrical stimulation: the early experiments. Phys Ther. 2002;82(10):1019-30. Epub 2002/09/28. PubMed PMID: 12350217.

Reviewer #3: Minor revisions:

1. The following prior comment was not adequately addressed. Indicate the underlying covariance structure used in the linear mixed model and the criteria for selecting it.

The top 3 most common covariance structures are compound symmetry, unstructured, and autoregressive.

Ａ：It is the linear mixed model using covariance structure of compound symmetry as the result that we showed it in the manuscript. We talked with a statistician. As you pointed it out, we tried the linear mixed model in 3 covariance structures (compound symmetry, unstructured, and autoregressive) with the statistician. The compound symmetry was the best as a result that it was judged in BIC (Bayesian information criterion). Therefore, the results do not change. Thus, we added the following sentences to the part of the statistical analysis.

“Also, we tried the linear mixed model in 3 covariance structures (compound symmetry, unstructured, and autoregressive). The compound symmetry was the best covariance structure as a result that it was judged in BIC. Therefore, we selected the compound symmetry as the underlying covariance structure used in the linear mixed model.” （P15, L234-238）

2- Table 1: Include the corresponding percentage male.

Ａ：I'm sorry, we were not able to revise it. We showed it in a table as follows. 

Male sex, n (%) 4 (80) 4 (80)

Also, I'm sorry, there were a few mistakes in the table. We revised diagonal characters and bold-face.

7. PLOS authors have the option to publish the peer review history of their article (what does this mean?). If published, this will include your full peer review and any attached files.

Do you want your identity to be public for this peer review? For information about this choice, including consent withdrawal, please see our Privacy Policy.

Reviewer #1: No

Reviewer #2: Yes: Bishnu Dutta Acharya

Reviewer #3: No

---

## [Decision Letter · Decision Letter 2]

28 Oct 2021

Electrically stimulated eccentric contraction during non-weight bearing knee bending exercise in the supine position increases oxygen uptake: a randomized, controlled, exploratory crossover trial

PONE-D-21-17794R2

Dear Dr. Matsuse,

We’re pleased to inform you that your manuscript has been judged scientifically suitable for publication and will be formally accepted for publication once it meets all outstanding technical requirements.

Kind regards,

Walid Kamal Abdelbasset, Ph.D.

Academic Editor

PLOS ONE

Additional Editor Comments (optional):

Reviewers' comments:

Reviewer's Responses to Questions

**Comments to the Author**

1. If the authors have adequately addressed your comments raised in a previous round of review and you feel that this manuscript is now acceptable for publication, you may indicate that here to bypass the “Comments to the Author” section, enter your conflict of interest statement in the “Confidential to Editor” section, and submit your "Accept" recommendation.

Reviewer #2: All comments have been addressed

Reviewer #3: All comments have been addressed

2. Is the manuscript technically sound, and do the data support the conclusions?

Reviewer #2: Yes

Reviewer #3: (No Response)

3. Has the statistical analysis been performed appropriately and rigorously? 

Reviewer #2: Yes

Reviewer #3: (No Response)

4. Have the authors made all data underlying the findings in their manuscript fully available?

Reviewer #2: Yes

Reviewer #3: (No Response)

5. Is the manuscript presented in an intelligible fashion and written in standard English?

Reviewer #2: Yes

Reviewer #3: (No Response)

6. Review Comments to the Author

Reviewer #2: 

Reviewer #3: (No Response)

7. PLOS authors have the option to publish the peer review history of their article (what does this mean?). If published, this will include your full peer review and any attached files.

Reviewer #2: No

Reviewer #3: No